# Modulating Kinetics of the Amyloid-Like Aggregation of *S. aureus* Phenol-Soluble Modulins by Changes in pH

**DOI:** 10.3390/microorganisms9010117

**Published:** 2021-01-07

**Authors:** Masihuz Zaman, Maria Andreasen

**Affiliations:** Department of Biomedicine, Aarhus University, 8000 Aarhus, Denmark; masih@biomed.au.dk

**Keywords:** functional amyloid, biofilm, phenol-soluble modulins, pH

## Abstract

The pathogen *Staphylococcus aureus* is recognized as one of the most frequent causes of biofilm-associated infections. The recently identified phenol-soluble modulin (PSM) peptides act as the key molecular effectors of staphylococcal biofilm maturation and promote the formation of an aggregated fibril structure. The aim of this study was to evaluate the effect of various pH values on the formation of functional amyloids of individual PSM peptides. Here, we combined a range of biophysical, chemical kinetics and microscopic techniques to address the structure and aggregation mechanism of individual PSMs under different conditions. We established that there is a pH-induced switch in PSM aggregation kinetics. Different lag times and growth of fibrils were observed, which indicates that there was no clear correlation between the rates of fibril elongation among different PSMs. This finding confirms that pH can modulate the aggregation properties of these peptides and suggest a deeper understanding of the formation of aggregates, which represents an important basis for strategies to interfere and might help in reducing the risk of biofilm-related infections.

## 1. Introduction

A range of microbes are known to form biofilms by assembling into well-organized and complex communities on surfaces to endure different environments [1,2]. Biofilm formation proceeds from the initial contact of an individual microorganism with a surface. An extracellular polymeric substance (EPS) composed of proteins, polysaccharides, lipids, and/or extracellular DNA (eDNA) is produced, and a complex, three-dimensional structure is established to form a mature biofilm [3]. The EPS maintains the structural integrity of biofilms, protects microbes from high doses of antibiotics and harmful conditions such as oxidative stress, along with attacks by other organisms (e.g., host immune system) [4,5]. It also promotes antibiotic tolerance in biofilms by stimulating the formation of a heterogeneous population of cells, due to the presence of environmental heterogeneity in the biofilm matrix itself [6]. Additionally, biofilms are known to affect wound healing in chronic wounds, and their power of eradication is significantly affected by changes in pH [7]. Aggregated proteins, in the form of functional bacterial amyloids, are key components of EPS, providing structural stability to biofilms [8,9]. The *Salmonella* Tafi protein, *Pseudomonas fluorescens* Fap proteins, *E. coli* curli, *Bacillus subtilis* TasA protein, and fungal hydrophobins are examples of such functional bacterial amyloids [10]. 

*S. aureus* is recognized as one of the most frequent causes of biofilm-associated infections [11]. The functional amyloid components of *S. aureus* are composed of small amphipathic peptides known as phenol soluble modulins [11,12]. PSM peptides are pronounced α-helical structures, which gives them surfactant-like characteristics, and are involved in a series of biological functions critical for staphylococci pathogenesis [13,14]. PSMs are classified as α-type (20–25 amino acids) and β-type (34–45 amino acids) according to their length. The PSMs of the α-type commonly have a net positive charge, except for δ-toxin, which possesses net neutral charge, while β-type PSMs have a net negative charge near physiological pH [15]. PSM peptide monomers can aggregate and form bacterial functional amyloids, which are considered to be associated with the stabilization of biofilm formation [16], with PSMα1 and PSMα4 aggregates displaying classical amyloid cross-β structures which are thermos-resistant [17], while PSMα3 aggregates display an unusual cross-α structure [18]. 

Despite the high sequence similarity between the PSM peptides, some of the peptides—namely PSMα2, PSMα4, and δ-toxin—do not aggregate at quiescent conditions without a facilitating agent [19]. As the aggregation is under kinetics control, a small change in the protonation states of key charged residues can lead to a dramatic change in the rate of fibril formation [20]. Additionally, an alteration in the protein sequence [21], addition of small molecules [22,23], and changes in pH have multiple effects on protein aggregation and biofilm formation [24,25]. Interestingly it has also been suggested that changes in the pH can act as a regulation mechanism for storage as amyloid structures and the subsequent release of peptide hormones have been observed in the pituitary gland [26]. Further, it has been shown that biofilm formation is largely affected by changes in pH in *Staphylococcus aureus* [27]. Enhanced fibril stability due to such changes can protect microorganisms against toxicity, by suppressing the development of toxic species, i.e., oligomers. Furthermore, differences in the in vivo environments where the functional amyloids form can alter the aggregation process; yet it remains unclear how localized cellular environments affect the dynamic equilibrium of aggregated fibrils. Hence, understanding the complex relationship of phenol soluble-modulins with biofilms entails a reflection on their structures and functions in various subcellular environments, beyond the cytoplasm.

In this study, we explored the effect of altering pH, focused on all seven PSM peptides, and dissected the propensity for individual monomers to form aggregated fibril structures. Using chemical kinetics, as well as biophysical and microscopic techniques, we demonstrate that despite their sequential and structural similarities, all peptides are highly sensitive to changes in pH and serve as an on/off switch for kinetics at different pH values. The results reveal how a subtle change in pH can amend the delicate kinetic balance and change the overall mechanistic character of the aggregation process of PSM peptides.

## 2. Materials and Methods

**Peptides, reagents, and solutions:** N-terminal modified (Formylation) PSM peptides at >95% purity were purchased from GenScript Biotech, Leiden, The Netherlands. Hexafluoroisopropanol (HFIP), trifluoroacetic acid (TFA), and Thioflavin T (ThT) were purchased from Sigma-Aldrich Ltd., St. Louis, MO, USA. Dimethyl-sulfoxide (DMSO) was purchased from Merck, Darmstadt, Germany. Ultra-pure water was used for the entire study. For measuring kinetics at different pH values, experiments were performed in the pH range of 1.0–12.0 with the following 20 mM buffers: pH 1 (KCl-HCl), pH 2 and pH 3 (Gly-HCl), pH 4 and pH 5 (Na-acetate-Glacial acetic acid), pH 6–8 (phosphate), pH 9–11 (Gly-NaOH) and pH 12 (KCl-NaOH). The ionic strength of the different pH buffers was calculated and found to be 0.10935 M (pH1), 0.0198 M (pH 2–3), 0.00168 M (pH 4–5), 0.03257 M (pH 6), 0.05598 M (pH 7), 0.09254 M (pH 8), 0.0198 M (pH 9–11), and 0.01935 M (pH 12), respectively. Buffers and stock solutions were filtered using PVDF 0.45 µm syringe filters (Millipore Milex-HV) before use.

**Peptide pre-treatment:** Each dry lyophilized PSM peptide stock was dissolved to a final concentration of 0.5 mg/mL in a 1:1 mixture of trifluoroacetic acid (TFA) and hexafluoroisopropanol (HFIP), followed by a 5 × 20 s sonication with 30 s intervals using a probe sonicator, and incubation at room temperature for one hour. Further, the stock solution was divided into aliquots, and the organic solvent was evaporated by speedvac at 1000 rpm for 3 h at room temperature. Dried peptide stocks were stored at −80 °C prior to use. 

**Preparation of samples for kinetic experiments:** ThT fluorescence was observed in clear-bottomed half-area 96 well black polystyrene microtiter plates (Corning, New York, NY, USA) with a non-binding surface. We used a Fluostar Omega (BMG Labtech, Orthenberg, Germany) plate reader in bottom reading mode. Aliquots of purified PSMs were thawed and dissolved in dimethyl sulfoxide (DMSO) to a concentration of 10 mg/mL prior to use. Concentrated peptide aliquots were diluted in MilliQ water and passed through a 0.22 μm filter. Freshly dissolved peptides were further diluted to the required concentrations into various buffers KCl-HCl for pH 1, Gly-HCl for pH 2–3, Na-acetate and Glacial acetic acid for pH 4–5, phosphate buffer for pH 6–8, Gly- NaOH for pH 9–11, and KCl-NaOH for pH 12. ThT was added to the protein solutions to a final concentration of 40 μM. Further, samples were loaded (100 μL) in a 96-well plate and sealed to prevent evaporation. For each peptide, different concentrations—0.25 mg/mL (PSMα1, PSMα2, PSMα4, PSMβ1 &2), 0.5 mg/mL (PSMα3), and 0.3 mg/mL (δ-toxin)—were used for kinetic measurements. The plates were incubated at 37 °C under quiescent conditions (i.e., absence of shaking and facilitating agents) and ThT fluorescence was measured every 10 min with an excitation filter of 450 nm and an emission filter of 482 nm. The ThT fluorescence was followed by three repeats of each monomeric concentration. The kinetic experiments with increasing peptide concentration were performed at 37 °C, monitored at every 10 minutes with an excitation filter of 450 nm and an emission filter of 482 nm under acidic and basic pH conditions. 

**Synchrotron radiation circular dichroism spectroscopy (SRCD):** The SRCD spectra of the various PSM fibrils were collected at the AU-CD beamline of the ASTRID2 synchrotron, Aarhus University, Denmark. To remove DMSO from the solution, fibrillated samples were centrifuged (13,000 rpm for 20 min), supernatants discarded, and the pellet resuspended in the same volume of milliQ water. Three to five successive scans over the wavelength range of 170 to 280 nm were recorded at 25 °C, using a 0.2 mm path length cuvette at 1 nm intervals with a dwell time of 2 s. All SRCD spectra were processed and subtracted from their respective averaged baseline (a solution containing all components of the sample, except the protein), smoothing with a 7 pt Savitzky-Golay filter, and expressing the final SRCD spectra in mean residual ellipticity. The SRCD spectra of the individual PSM fibrils samples were deconvoluted using DichroWeb (Department of Crystallography, Institute of Structural and Molecular Biology, Birkbeck College, Unversity of London, UK) [28,29] to obtain the contribution of individual structural components. Each spectrum was fitted using the analysis programs Selecon3, Contin, and CDSSTR with the SP175 reference data set [30], and an average of the structural component contributions from the three analysis programs was used. 

**Circular dichroism analysis:** Far-UV CD spectra were measured in a JASCO-810 (Jasco Spectroscopic Co., Ltd., Hachioji City, Japan) spectrophotometer equipped with a Peltier thermally controlled cuvette holder at 25 °C. Aggregated peptides were centrifuged and traces of DMSO were removed to avoid any noise in the CD signal. Spectra were obtained by averaging five individuals spectra recorded between 250 and 190 nm, at 0.1 nm intervals, 1 nm bandwidth, a scan speed of 100 nm/min, and a response time of 1 s. Spectra of the buffers’ only controls were subtracted from the spectra of the proteins.

**Fourier Transform Infrared (FTIR) Spectroscopy:** Fourier transform infrared spectroscopy was recorded on a Tensor 27 FTIR instrument (Bruker Optics, Billerica, MA, USA) equipped with an attenuated total reflection accessory with a continuous flow of N_2_ gas. To remove DMSO from the solution, fibrillated samples were centrifuged (13,000 rpm for 20 min), supernatants discarded, and the pellet resuspended in MilliQ water. Prior to measurement, 5 μL samples were dried under a stream of N_2_ gas, and 64 interferograms were accumulated with a spectral resolution of 2 cm^−1^ in the range of 1000–3998 cm^−1^. OPUS 5.5 software (Bruker Optics, Billerica, MA, USA) was used to process the data, which included baseline subtraction, atmospheric compensation, and a second derivative analysis. For comparative studies, all absorbance spectra were normalized. Only the range of 1600–1700 cm^−1^ comprising information about the secondary structure is shown. 

**ANS fluorescence:** ANS fluorescence spectra were measured on an LS55 luminescence spectrophotometer (Perkin Elmer Instruments, Waltham, MA, USA) using excitation and emission slit widths of 2.5 nm. The experiments were performed in a black 3 mm × 10 mm quartz cuvette at 25 °C. A hydrophobic dye 1-anilino-8-naphthalene sulfonate (ANS) was dissolved in MilliQ water, filtered with a 0.45 µm filter. Furthermore, its concentration was determined with Nanodrop (Thermo Scientific, Waltham, MA, USA) using a molar extinction coefficient of 4990 M^−1^ cm^−1^ at 350 nm [31]. Fibrils of different peptides at different pH values were prepared, incubated with ANS, and kept for 30 min in the dark. Excitation was performed at 380 nm with emission spectra recorded from 400–600 nm. Two scans were averaged using a step size of 1 nm and an integration time of 0.1 s. 

**Transmission electron microscopy:** For the TEM analysis, each PSM (one acidic and one basic pH) peptide sample (5 μL) was deposited onto carbon coated formvar EM grids (EM Resolutions, Sheffield, UK) and incubated for 120 s at room temperature. The buffer was subsequently removed from the grid by blotting the grid into Whatman filter paper. Further, the grid was washed with MilliQ water (5 μL) and stained with 1% uranyl acetate for 120 seconds, and the excess of staining solution was removed from the grid with Whatman filter paper. Finally, the grid was washed twice with a drop of MilliQ and blotted dry on filter paper for 10 minutes. Images were recorded using Morgagni 268 (FEI Phillips Electron microscopy, Hillsboro, OR, USA), equipped with a CCD digital camera, and operated at an accelerating voltage of 80 kV. 

## 3. Results

### 3.1. Aggregation of PSM Peptides at Acidic pH

To investigate how different pH values alter the kinetic behavior of PSM peptides, monomers of all seven PSM peptides (PSMα1-4, PSMβ1-2, and δ-toxin) were incubated at a pH range from 1 to 12, and the aggregation kinetic was monitored using ThT. Fibril formation processes generally exhibit a sigmoidal kinetic behavior characterized by a lag-phase, growth, and a stationary phase [32], and different solution conditions such as a change in salt concentration and pH can induce different strains of fibrils, resulting in polymorphism. At acidic pH from pH 1 to pH 4, PSMα1 forms fibrils within 10 to 40 h that have a significant ThT fluorescence signal and a half-time that decreases with increased pH values (Figure 1a). At pH 5 and 6 no aggregation is observed, while at pH 7 sigmoidal aggregation kinetics are restored but with an increase in half-time compared to lower pH values (Figure 1a). Consistent with previous reports, no aggregation was observed for PSMα2 at pH 7 [19], but this is also the case at all the acidic pH values tested here (Figure 1b and Figure 2b). For PSMα3, no aggregation occurs at acidic pH; only upon reaching pH 7 does the peptide display sigmoidal aggregation, as also reported previously (Figure 1c). Even after prolonged incubation, no aggregation occurs at acidic pH (Appendix A). For PSMα4, aggregation is observed at extreme acidic pH values (pH 1–3), whereas at mild acidic pH (4–6) and at neutral pH no aggregation is observed (Figure 1d). At extreme acidic pH values (pH 1 and 2), and again at neutral pH values, PSMβ1 aggregates, although with different kinetic curve shapes (Figure 1e). At neutral pH, PSMβ1 displays a standard sigmoidal aggregation kinetic curve, but at low pH the aggregation is seen as an immediate rise in the ThT signal with no significant lag-time, reminiscent of seeded aggregation kinetics. In the intermediate pH values (pH 3–6), no aggregation is observed (Figure 1e). For PSMβ2, no aggregation is seen at acidic pH (Figure 1f). Only at pH 7 does aggregation occur. This behavior is very similar to that of PSMα3 at acidic pH. For the last PSM peptide, δ-toxin aggregation is only observed at more extreme pH values (pH 1–3), with increasing half-times with increased pH values (Figure 1g), and thus displaying a behavior very similar to that of PSMα4. Based on the aggregation behavior at increased pH values, the PSM peptides can be divided into three different groups. The first group consists of PSMα1 and PSMβ1, where aggregation is seen at extreme acidic pH values, followed by no aggregation at mildly acidic pH, only for aggregation to resume at neutral pH. The second group, where no aggregation is seen at any acidic pH values but occurs at pH 7, consists of PSMα3 and PSMβ2. PSMα2 could also be considered to be part of this group, but with no aggregation also occurring at pH 7. The last group only displays aggregation at extreme acidic pH values and consists of PSMα4 and δ-toxin. When aggregation occurs at acidic pH values, the general trend observed is that the half-time increases with increasing pH values.

The isoelectric point for all PSMα’s and δ-toxin is above pH 7, and hence changes in the aggregation behavior at acidic pH are not caused by changes in the charge of the peptide from positive to negative. For PSMβ2, this switch in the aggregation behavior coincides with the introduction of an overall negative charge by pH values above the pI of PSMβ2 (pI of 5.69). However, for PSMβ1 the changes in aggregation behavior do not coincide with the pI of the peptide of 4.89, as aggregation occurs at pH 1, 2, and 7. Hence, the switch from an overall positive charge to an overall negative charge is not the sole reason for the changes in aggregation propensity.

### 3.2. Aggregation of PSM Peptides at Basic pH

At basic pH, several of the PSM peptides show a lack of ThT fluorescence. For PSMα2, PSMα4, and δ-toxin, no significant increase in ThT fluorescence is observed at any of the basic pH values tested here (Figure 2). Besides the aggregation curves observed at pH 7, PSMα1 also display aggregation curves at pH 9 and 10 but not at pH 8 or the more extreme basic pH values of 11 and 12. The fastest aggregating PSM, namely PSMα3, aggregates at all basic pH values except for pH 9, and hence aggregates at both mildly basic and more extreme basic pH values, although the aggregation at more extreme pH values occurs with a significantly longer lag-time compared to neutral pH and pH 8. For PSMβ1, aggregation is observed from pH 7 to pH 10 but not at pH 11 and 12 (Figure 2e), whereas PSMβ2 displays an increase in ThT fluorescence at all basic pH values from pH 8 to pH 12, hence indicating that aggregation occurs from pH 7 to pH 12 (Figure 2f).

For PSMα3, the lack of aggregation at pH 9 does not exactly coincide with the pI of 9.53, but does coincide with the pH value with the lowest numerical value of the charge of the peptide, and hence can be linked to the charge status of the peptide. This is not the case with the PSMα1with a pI of 9.72, which, contrary to PSMα3, coincides with the aggregation curve seen at both pH 9 and pH 10. Moreover, we observed significant differences in ThT fluorescence intensity between all the peptides at the said conditions, which might be due to differences in the degree of formation of aggregates that bind to ThT. Further, the morphological analysis (discussed later) surprisingly indicated that δ-toxin forms aggregates irrespective of the insignificant fluorescence intensity, as observed by ThT kinetics (Figure 2g) at basic conditions (pH 9–12). Irrespective of their aggregation kinetics for their fibril formation, these results show that changes in the pH by one unit alter the kinetics mechanism of fibril assembly. From these results, we can say that the acidic and basic residue present in the sequence of peptides may change the ionization state between pH 1 and pH 12, and may be responsible for the fibril formation, as reported earlier for the semen cationic peptide responsible for HIV infection [33]. Further, the net charge (calculated by protein calculator v3.4) developed in all these peptides at different pH values (Appendix A) might also play an important role in determining the propensity toward aggregation, as observed earlier for GLP-1 protein [34,35].

### 3.3. Effect of Peptide Concentration

Previously, it has been shown that changes in the protein concentration can lead to fibrillar polymorphism due to an increased chance of association at increased concentrations (due to solubility limits), as well as a decreased aggregation due to the crowding effect [36,37]. Using a chemical kinetic analysis of aggregation of PSM peptides at different concentrations at neutral pH, we have previously shown that PSMα1, PSMα3, and PSMβ1 aggregate through a secondary nucleation dominated mechanism, while PSMβ2 aggregation is dominated by primary nucleation and elongation with absence of secondary processes. In order to investigate the effects of the changes in the protein concentration on the aggregation of PSM peptides at pH values different from pH 7, the aggregation kinetics of different PSM peptides were monitored at selected pH values. The aggregation kinetics of PSMα1 (0.25–0.62 mg/mL) was monitored at pH 2, PSMβ1 and PSMβ2 (0.25–0.62 mg/mL) were monitored at pH 10, and δ-toxin (0.30–0.75 mg/mL) was monitored at pH 2 using ThT fluorescence. The results are summarized in Figure 3. Upon incubation under quiescent conditions, standard sigmoidal curves with increased ThT fluorescence intensity were obtained for all the concentrations of these PSM peptides (Figure 3a–d), but the kinetic parameters and helf-time obtained from the fitting differ with the pH and peptide concentration, as summarized in Appendix A. Usually, with increasing peptide concentration, an increase in the rate of fibrillation with reduced half-times can be observed, a feature that is characteristic of a nucleation-polymerization mechanism [38]. However, at lower pH values (pH 2), there is either very little dependence on the half-time of peptide concentration or even an increase in *t*_1/2_ with increasing peptide concentrations, as observed for PSMα1 and δ-toxin (Appendix A). At basic pH (pH 10), the *t*_1/2_ of PSMβ2 was found to decrease with increasing peptide concentrations, characteristic of a process following a nucleation-polymerization mechanism for fibril formation (Appendix A). However, for PSMβ1, which belongs to the same group of βPSMs, *t*_1/2_ increases with increasing peptide concentrations at higher pH values, as observed for the two other peptides, i.e., PSMα1 and δ-toxin (Appendix A). These observed increases in *t*_1/2_ with increasing peptide concentrations suggest that another process comes into play (the opposite of what was expected for an N-P model), as observed for other systems [39,40]. This could also be related to saturation effects during the aggregation reaction. However, these increments are not as high as the ones observed for the GLP-1 protein near the physiological pH [35]. 

### 3.4. Structural Characterization of PSM Aggregation In Vitro

The conformational changes of all PSM peptides following aggregation at acidic and basic conditions were monitored using synchrotron radiation circular dichroism (SRCD) spectroscopy and Fourier transform infrared (FTIR) spectroscopy. The CD spectrum of the native peptide of all PSMs at the beginning of the time course shows α-helical characteristics with a double minimum at 208 and 222 nm (Appendix A). Despite the common α-helical starting point, the spectra recorded at different pH values are slightly different, displaying different minima. A predominant minimum at approximately 218 nm (PSMα1, PSMα4 and δ-toxin) and 220 nm (PSMβ1) was observed for solutions of aggregated peptides at pH 2 (Figure 4a). This indicates a transition from an α-helical structure to a structure with increased β-sheet content upon aggregation, which is consistent with data previously published [2,41]. However, no apparent conversion was observed for PSMα2, PSMα3, and PSMβ2 peptides (Appendix A) at acidic conditions (pH 2), which is consistent with the lack of aggregation seen for these peptides by ThT fluorescence. At basic conditions (pH 10), a single negative peak at around 220 nm was observed for PSMα1, PSMα4, PSMβ1, and PSMβ2 (Figure 4b). The structural conversion of these peptides into a β-sheet signal is in good agreement with our kinetics data, where aggregation was observed for the peptides at pH 10. The CD spectrum of aggregated PSMα3 and δ-toxin at pH 10 still display a double minimum with minima shifted to 208 nm and 228 nm, indicative of α-helical structure being present in the aggregates (Figure 4b). The helical structure of aggregated PSMα3 is different from that observed in the monomeric peptide and is consistent with the reported cross-α-helical-like structure of PSMα3 aggregates [18]; however, the aggregated δ-toxin’s helical structure is almost similar to the native one. 

Interestingly, FTIR results indicate significant differences in the secondary structure content as calculated by the deconvolution of the absorbance spectra in the amide I´ region at different pH values. This approach allows for an analysis of the individual secondary structure components and their relative contribution in the main signal [42]. The results are summarized in Appendix A. The intense signal around 1625 cm^−1^ for PSMα1, PSMα4, PSMβ1, and δ-toxin is indicative of intermolecular β-sheets packed into the characteristic amyloid fibrils structure [43] and is consistent with our CD data. However, PSMβ1 also show a signal at ~1664 cm^−1^ corresponding to β-turn conformations [42]. In contrast, at basic pH, all peptides except PSMα2, PSMα3, and δ-toxin display a common band at ~1625 cm^−1^, attributed to intermolecular amyloid-like β-sheets and a band at ~1645 cm^−1^, corresponding to a random coil conformation [42]. However, a band at ~1645 cm^−1^ by PSMα2 and δ-toxin, typically assigned to a helical/random conformation, arises at basic conditions. This may reflect an equilibrium between residual helical soluble states and a predominant aggregated assembly. However, as the FTIR analysis is carried out on a pelleted sample, the contribution from the soluble peptide is not expected to be significant. The characteristic high frequency band at ~1625 cm^−1^ region suggests a high content of calculated β-sheet (~71%) for PSMα1 in an acidic pH range, whereas in basic conditions, the peak was less sharp, and the calculated β-sheet content was found to amount to ~48%. Additionally, PSMα3 did not experience any significant conformational change during the same incubation period in acidic conditions (Appendix A). However, at basic conditions a more intense band in the spectrum of ~1655 cm^−1^ leads to the presence of ~80% α-helical-like conformations, as it maintains its α-helical conformation in both conditions, i.e., in solution and in the fibril form, as confirmed by the X-ray diffraction pattern [18]. The percentage of β-sheet content of PSMα4 in acidic conditions was slightly higher than that at basic pH, as calculated by the FTIR spectrum (Appendix A). The presence of ~20% high frequency β-sheet signal at 1690 cm^−1^, characteristic of an anti-parallel β-sheet, suggests that in PSMα4 peptides, the β-sheet peptides are packed in the fibrils in anti-parallel fashion too [11]. In contrast, the conversion of PSMβ1 and PSMβ2 into a predominant β-sheet structure revealed that the secondary structure of the aggregates is very similar across the basic pH and possesses an almost similar percentage (~45%) of β-sheet structures. For all the peptides analyzed at basic pH, a peak is also observed at ~1606 cm^−1^. This peak can be assigned to the glycine molecules of the buffer, as this band has previously been assigned to glycine [44]. These results are in agreement with the data obtained by the CD data of the peptides under different conditions.

### 3.5. Characterization of the Hydrophobicity of the PSM Aggregates

The solvent-exposed hydrophobic surface area of the aggregates formed at pH 2 and pH 10 was probed using 8-Anilino-1-naphthalenesulfonic acid (ANS) [45]. ANS binding was assessed to monitor the alteration of the microenvironment using fluorescence measurements over a fixed peptide concentration. A significant increase in ANS fluorescence intensity at pH 2 and pH 10 indicates that PSMs are highly prone to form aggregates with an exposed hydrophobic surface area (Appendix A). However, all peptides showed different ANS fluorescence intensities and λ_max_ peak positions at both acidic and basic conditions. The maximum fluorescence intensity at acidic conditions was found for PSMα4 with respect to PSMα1, δ-toxin, and PSMβ1 (Figure 4e). Additionally, the decrease in the ANS fluorescence intensity also reveals that the size of the particles varies, as the ANS fluorescence intensity varies with the size of aggregated species [46]. Moreover, we also observed a spectral blue shift with preferential binding of ANS (Figure 4e). This may be due to a change in the intramolecular charge transfer rate by the positive charge that is present near the –NH group of ANS [34]. However, no significant fluorescence intensity was observed for the rest of the peptides (PSMα2, PSMα3, and PSMβ2) at these conditions (acidic pH), confirming the lack of the hydrophobic patches that are typically present in aggregates or aggregation-prone peptides (Figure 4e). Similar observations were obtained with ANS fluorescence at basic conditions of the PSM peptides, and the maximum ANS fluorescence intensity was observed for PSMα1 with respect to other peptides (Figure 4f). Further, PSMβ1 and PSMβ2 did not show any significant shift in blue spectral lines, as observed in acidic conditions for PSMβ1 (Figure 4f). However, at basic pH (pH 10), the reduction in ANS fluorescence intensity by δ-toxin clearly indicates that less hydrophobic patches were exposed to the dye. A remarkable spectral blue shift of up to 19 nm by this peptide at basic pH might be due to the presence of a non-polar environment to the exposed Trp residues inside the aggregates (Figure 4f). Overall, we observed a pH-dependent change in ANS fluorescence intensity due to a variation in the exposed hydrophobic surface area of the PSM aggregates at various solution conditions.

### 3.6. Morphological Characterization of PSM Aggregates

The macromolecular morphological structures of PSM peptide solutions were examined by transmission electron microscopy (TEM). In good agreement with the recorded aggregation kinetics and structural data, ordered fibrils were observed for some PSM peptides at acidic (pH 2) and basic (pH 10) conditions (Figure 5). The PSMα1 (Figure 5a), PSMα4 (Figure 5c), PSMβ1 (Figure 5d), and δ-toxin (Figure 5k) solutions exhibited a large number of unbranched, thin, and long amyloid-like fibrils at pH 2. In the case of PSMα4 and PSMβ1, a heterogeneous species appeared, and they seem to consist of both fibrillar and amorphous aggregates (Figure 5c,d). However, the distribution, structure, and size of δ-toxin fibrils at acidic conditions are unique and very different from the other peptides (Figure 5k). Conversely, PSM solutions exhibited a small number of amorphous aggregates for PSMα3 (Figure 5b), short protofibrillar structures for PSMβ2 (Figure 5i), or the absence of fibrillar structures for PSMα2 (Appendix A) at pH 2. Replacing the solution conditions with basic pH PSM peptides results in the formation of fibrils. However, the resulting aggregates did not resemble each other, and differences in fibril morphologies were observed. At basic pH, PSMα1 forms long and homogenous fibrils that have less fibrils as compared to acidic pH (Figure 5e). The observed compact and dense fibril structure of PSMα3 at basic pH suggests that it is a significant contributor to fibrillation at basic pH (Figure 5f). However, as observed in acidic pH, increased fibril heterogeneity appears for PSMα4, containing both fibrillar and amorphous structures of varying size and shapes (Figure 5g). In contrast to the PSMα groups, at basic pH, our TEM images show that there is a conversion of the monomeric form to the aggregated form by both peptides of the β-group, i.e., PSMβ1 (Figure 5h) and PSMβ2 (Figure 5j). Further, at basic pH, a more homogeneous population was observed for PSMβ1 as compared to that seen at acidic pH. More strikingly, we note that δ-toxin exhibits a low ThT signal that increases modestly upon 30 h of incubation, suggesting that while the population of fibrils is less dense, the mature aggregates do contain some amyloid-like structures (Figure 5l). Lastly, some rod-shaped structures were observed for PSMα2 samples when incubated at pH 10 for four days (Appendix A). PSMα1, PSMα4, PSMβ1, and δ-toxin, on the other hand, are necessary and sufficient for fibrillation at both pH ranges, i.e., acidic and basic, as confirmed by chemical kinetics studies, secondary structural analysis, and morphological analysis.

## 4. Discussion

PSM peptides are key determinants of *Staphylococcus aureus* virulence [14]. Among them the α-group of PSMs are the smallest staphylococcal toxins and have been well characterized due to their immune modulating and their cytolytic activity [47,48]. In recent times, various studies have described PSMs’ participation in biofilm formation and detachment, suggesting new PSM functions in the staphylococcal pathogenesis [49,50]. Remarkably, PSMs appear to act by forming fibrillar amyloid-like structures that ensure the integrity of the *S. aureus* biofilms [13,16]. In order to understand the complex, pH-dependent aggregation, we analyzed *S. aureus* PSMs’ (all seven) amyloid formation mechanism using a range of pH (1–12) to dissect the contribution of these short peptides to the biofilm structure. The extracellular matrix of *S. aureus* is highly complex, so the interaction between PSMs peptides with other matrices or matrix-like components, e.g., eDNA, heparin, exopolysaccharides, and alginate, must be considered to completely reveal the effect of pH on aggregation under in vitro as well as in vivo conditions. Our present study of PSM behavior with different pH values was intended to mimic variable biofilm conditions, and we confirmed that all PSMs fibrillate to a greater extent at moderate and high pH values. However, some of them also tend to fibrillate at low pH values. This indicates that the presence of different PSM peptides ensures the formation of aggregated functional amyloids at a large range of pH values that could be encountered by the bacteria during biofilm formation in vivo.

Earlier, computational and experimental analyses already suggested that the peptides in this family might exhibit differential self-assembly properties [11]. These were confirmed by in vitro experiments [19], demonstrating that not all PSMs form aggregated structures as previously thought [16]. According to our findings, PSMα1, PSMα4, δ-toxin, and, to some extent, PSMβ1 peptides would be major contributors to PSM fibrillogenesis at acidic pH (Figure 1). The structural reorganization at low pH can be simplified by considering the distribution of charges that exist in the low pH form relative to the neutral pH, as observed earlier for α-synuclein [51]. Amyloid fibrils, including functional amyloids from *Pseudomonas*, are highly resistant to acidic conditions [52], so increased fibril formation at low pH values could help to provide a dense mesh of protective proteins to shield the bacteria in the biofilm from harsh conditions.

Despite the high sequence similarity between PSMα1 and PSMα2, the fibrillation differences between these two peptides are distinct. The sequences of all peptides are summarized in Appendix A. This might be due to the presence of a discrete number of total charges and pK_a_ (presence of different amino acid as well as charges), that plays an important role in determining its propensity towards aggregation [34,35]. This pH-dependent pK_a_ shift is sensible, given that it would be more favorable to have neutral rather than charged groups buried within the hydrophobic core, and hence protonation is facilitated upon aggregation [53]. Interestingly, based on our results, in contrast to acidic pH, biofilm structuring in basic pH solutions is completed by almost every PSM peptide that effectively shows the formation of aggregated fibrils with differences in the aggregation kinetics, morphology, and extent of β-sheet formation. Importantly, our data also indicate that despite a higher pH exerting an inhibitory effect on the adhesion of *S. aureus* with impairment of the maturation of biofilms [25], PSM peptides formed fibrils at higher pH values and play an important role in the structural integrity of biofilms. In addition, at physiological pH it is the βPSM group of peptides, along with PSMα3, which forms aggregates and plays a significant role in the structuring of *S. aureus* biofilms [16]. 

PSM peptides are not only related to the formation of amyloid-like fibrils in the biofilm, but also to biofilm disassembly [13,54,55], lytic activities against host cells [16], and antimicrobial activities against other microorganisms [56,57,58], also acting as virulence factors [59,60,61]. These other activities of peptides are also likely affected by changes in the pH, as changes to the protonation state of the peptides not only affect the aggregation properties but also, e.g., the propensity to form helical structures. It has been shown that the fibrillation of PSMα3 increases the toxicity against human cells [18], and hence changes in the aggregation behavior of PSMα3 due to changes in the pH will also affect the cytotoxicity of this peptide. Furthermore, PSM peptides have been found to undergo truncations in vivo, leading to new functionalities, e.g., antimicrobial activities, as a response to external stimuli [62,63,64,65]. The truncation of the peptides could also be affected by the changes in the pH of the surrounding environment, and hence affect the antimicrobial activities against other microorganisms.

The secretion of all *S. aureus* PSMs are strongly dependent on pmt (PSM transporter) and provide a promising therapeutic target, because their inactivation would avert the translocation of PSM peptides to the extracellular space [66]. This leads to their cytoplasmic accumulation, resulting in a loss of bacterial fitness [66]. Our results show that near the physiological pH, the deleterious impact of the intracellular accumulation of PSMs in *S. aureus* upon the blockage of their secretion would be essentially exerted by more soluble PSMs (αPSMs), and not by βPSMs that would likely aggregate into inert inclusions. Interestingly, when PSMs that accumulated in the cytosol of a pmt deletion mutant were analyzed, only αPSMs could be identified but not βPSMs, i.e., PSMβ1 and PSMβ2 [11,66]. It was projected that this nonexistence would be due to degradation, non-specific adhesion to cellular material, or lower production, but our outcomes sturdily suggest that they were not identified due to the accumulation as insoluble aggregates. Therefore, they were not present in the analyzed cellular supernatant [11,66]. However, there might be similar processes occurring at acidic and basic pH solutions where some of the PSMs that tend to aggregate are not identified, while the rest (non-aggregated) are found in the cellular supernatant. Importantly, our data also indicate that the α-helical propensities of these peptides might differ significantly with the pH values encountered and is likely an important factor to determine the aggregation potential. The lytic activity of PSMs that share a conserved α-helical structure is supposed to be somehow associated with the hydrophobic character of the α-helical motif, since hydrophobic residues are expected to promote PSM aggregation, concurrently reducing the quantity of active available peptides. This will decrease the peptides’ interactions with membranes and their disruptive effect [67,68] in a mechanism analogous to that described for the α-helical antimicrobial peptides [69,70]. Marinelli et al. calculated the hydropathy index of the α-PSM group of peptides and reported that the hydrophobic score was found to be maximum for PSMα4, while PSMα3 was found to be the least hydrophobic of the peptides [11]. These values are in good agreement with our ANS binding data (acidic pH), as a maximum was found for PSMα4, followed by PSMα1 and PSMα3, with their distinct functions. This could suggest that a different hydropathy index, along with the spatial distribution of the residues in the different PSMs peptides of the α-group of PSMs, contributes to their different aggregation propensity [11]. Additionally, the change in amplitude in blue shifts by PSMs at different pH also suggests a different fibril polymorph, which has been observed in earlier studies for other amyloidogenic proteins as a function of pH [71,72]. Furthermore, the spectral blue shift (maximum in δ-toxin at alkaline condition) suggests that it is possible that alkaline pH may directly influence the physical and/or chemical surface properties of bacterial peptide results in altering the peptide conformation [25] thought to govern surface hydrophobicity. Therefore, they have an additional role in peptide aggregation.

In conclusion, we show that aggregation kinetics was largely exaggerated by solution conditions and confirms the formation of aggregates in PSMs that are responsible for biofilm formation in *S. aureus*. These results indicate the likely existence of different pathways in fibril formation and serve to shed light on mechanisms of aggregation in these complex peptides. Most importantly, they will provide new targets that can be exploited for therapeutics, as PSMs that possess a critical and diverse role during infection signify a promising target for anti-staphylococcal therapy [15]. Together, we also found a vast structural diversity of amyloid-like structures in *S. aureus* peptides that may generate various virulence activities encoded by diverse aggregate morphologies when secreted concurrently at numerous subcellular surroundings. These insights are highly valuable in the development of antibiotic alternatives for combating biofilms and ultimately reducing the burden of hospital-acquired infections. However, more research is required to evaluate the role of solution conditions along with other extracellular matrix components, to clarify how these effects contribute to the formation and stabilization of biofilm formation of *S. aureus* under in vitro as well as in vivo conditions.

## Figures and Tables

**Figure 1 microorganisms-09-00117-f001:**
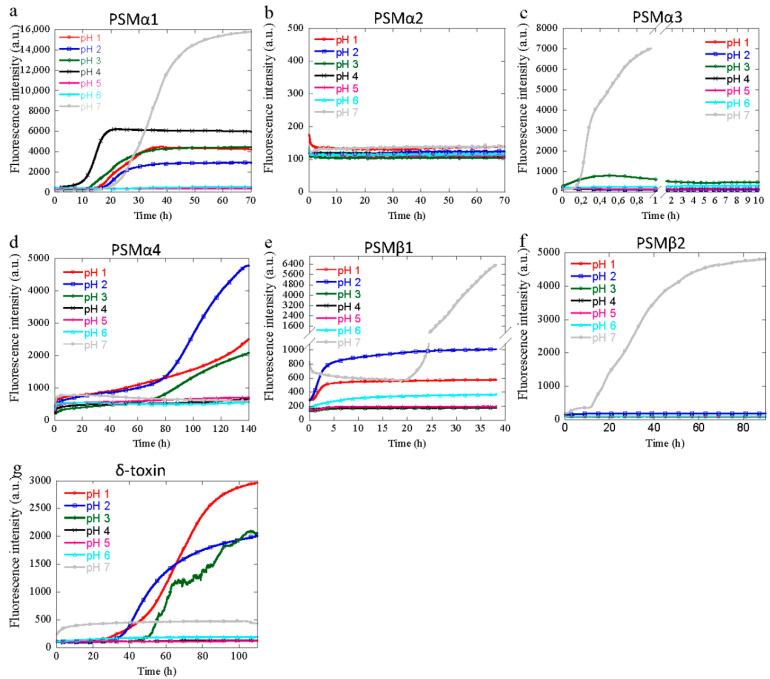
Experimental kinetics of PSM aggregation at 37 °C every 10 min under quiescent conditions for single monomeric concentration at acidic and neutral (pH 1–7) solution conditions followed by ThT fluorescence. Three repeats were carried out at each condition. (**a**) Aggregation kinetics of PSMα1 (0.25 mg/mL). (**b**) Aggregation kinetics of PSMα2 (0.25 mg/mL). (**c**) Aggregation kinetics of PSMα3 (0.5 mg/ml). (**d**) Aggregation kinetics of PSMα4 (0.25 mg/mL). (**e**) Aggregation kinetics of PSMβ1 (0.25 mg/mL). (**f**) Aggregation kinetics of PSMβ2 (0.25 mg/mL). (**g**) Aggregation kinetics of δ-toxin (0.3 mg/mL).

**Figure 2 microorganisms-09-00117-f002:**
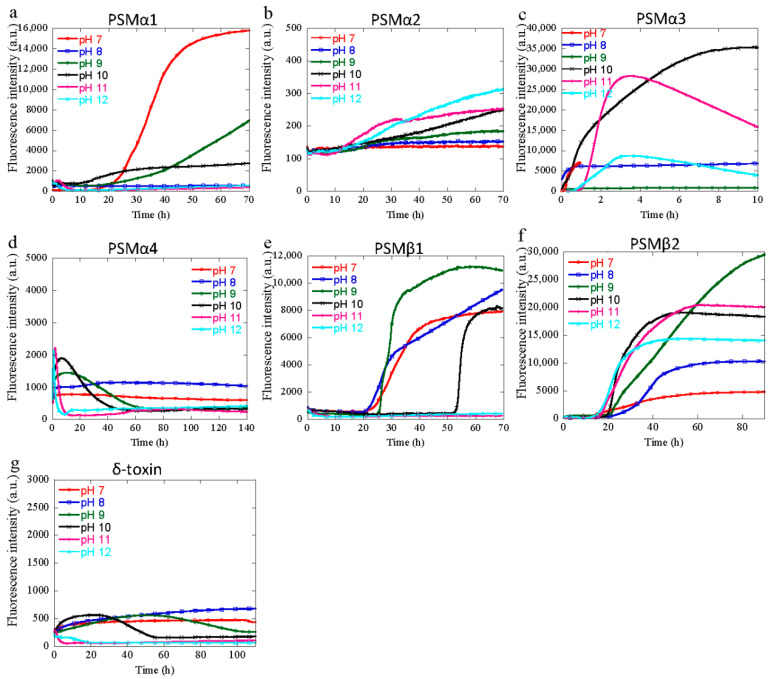
Experimental kinetics of PSM aggregation at 37 °C every 10 min under quiescent conditions for single monomeric concentration at basic and neutral (pH 7–12) solution conditions. Three repeats were carried out at each conditions. (**a**) Aggregation kinetics of PSMα1. (**b**) Aggregation kinetics of PSMα2 (0.25 mg/mL). (**c**) Aggregation kinetics of PSMα3 (0.5 mg/mL). (**d**) Aggregation kinetics of PSMα4 (0.25 mg/mL). (**e**) Aggregation kinetics of PSMβ1 (0.25 mg/mL). (**f**) Aggregation kinetics of PSMβ2 (0.25 mg/mL). (**g**) Aggregation kinetics of δ-toxin (0.3 mg/mL).

**Figure 3 microorganisms-09-00117-f003:**
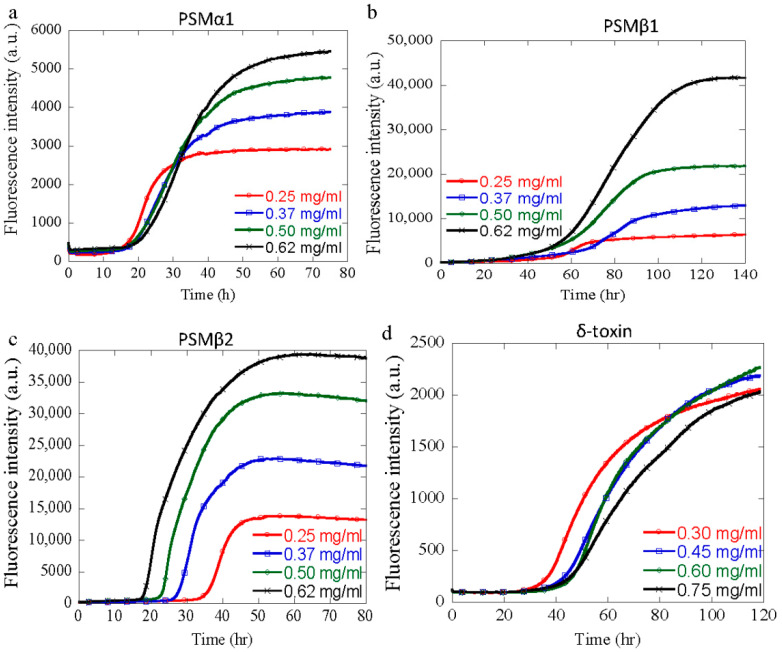
Aggregation kinetics of different PSMs as a function of peptide concentration at fixed pH. Each peptide concentration was run in triplicates. (**a**) Aggregation kinetics of PSMα1 at pH 2 show fibrillation with lag-phase followed by ThT fluorescence at various concentrations (0.25–0.62 mg/mL). (**b**) Aggregation kinetics of PSMβ1 at pH 10 show fibrillation with lag-phase followed by ThT fluorescence at various concentrations (0.25–0.62 mg/mL). (**c**) Aggregation kinetics of PSMβ2 at pH 10 show fibrillation with lag-phase followed by ThT fluorescence at various concentrations (0.25–0.62 mg/mL). (**d**) Aggregation kinetics of δ-toxin at pH 2 show fibrillation with lag-phase followed by ThT fluorescence at various concentrations (0.3–0.75 mg/mL).

**Figure 4 microorganisms-09-00117-f004:**
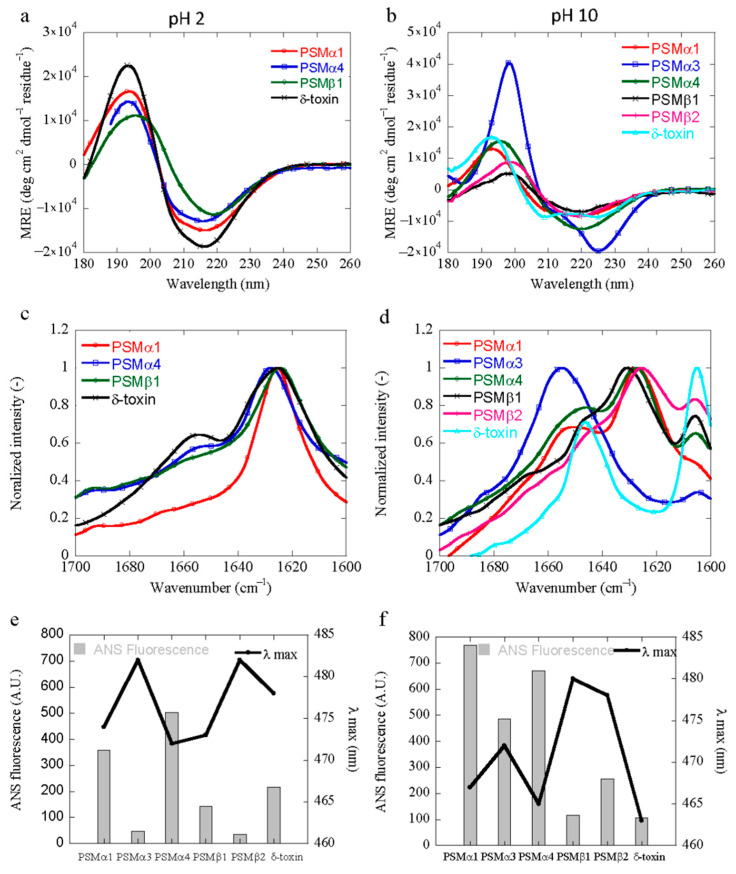
Biophysical structural characterization of PSM fibrils formed under various solution conditions (**a**) Synchrotron radiation CD (SRCD) spectra of different PSMs at acidic pH (pH 2). Spectra of different peptides acquired at the same pH show different minima, and the peak positions are different from each other. (**b**) SRCD spectra of different PSMs at basic pH (pH 10). The variation was modest, as observed at acidic conditions, but almost all peptides show single negative minima. (**c**) FTIR spectroscopy of the amide I’ region (1600–1700 cm^−1^) of fibrils of PSM variants at acidic (pH 2) solution conditions. PSMα1, PSMα4, PSMβ1, and δ-toxin show a peak at 1625 cm^−1^ corresponding to rigid intermolecular β-sheet structures, as seen in amyloid fibrils. (**d**) FTIR spectroscopy of the amide I’ region (1600–1700 cm^−1^) of fibrils of PSMs variants at basic solution (pH 10) conditions. All peptides except δ-toxin show a peak around 1625 cm^−1^ corresponding to intermolecular β-sheet fibrillary structures. In contrast, δ-toxin shows two peaks around peaks at and around 1645 cm^−1^, with the latter indicating more disordered fibrils. In contrast, PSMα3 shows main peaks at and around 1654 cm^−1^, with the latter indicating more disordered fibrils. (**e**) ANS fluorescence intensity and changes in emission maxima (λ_max_) of PSMs at acidic (pH 2) conditions with fixed peptide concentrations. (**f**) ANS fluorescence intensity and changes in emission maxima (λ_max_) of PSMs at basic (pH 10) conditions with fixed peptide concentrations.

**Figure 5 microorganisms-09-00117-f005:**
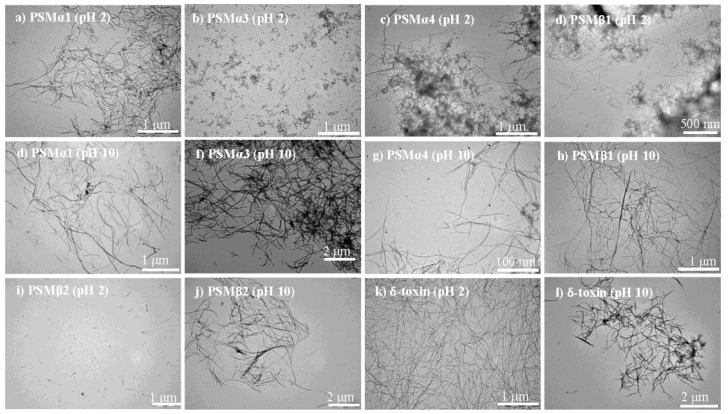
Morphology of aggregates of PSM peptides at one acidic (pH 2) and one basic (pH 10) condition. Transmission electron microscopic image of the end-state of reaction for samples initially composed of (**a**) PSMα1 fibrils (pH 2), (**b**) PSMα3 fibrils (pH 2), (**c**) PSMα4 fibrils (pH 2), (**d**) PSMβ1 fibrils (pH 2), (**e**) PSMα1 fibrils (pH 10), (**f**) PSMα3 fibrils (pH 10), (**g**) PSMα4 fibrils (pH 10), (**h**) PSMβ1 fibrils (pH 10), (**i**) PSMβ2 fibrils (pH 10), (**j**) PSMβ2 fibrils (pH 2), (**k**) δ-toxin fibrils (pH 2), and (**l**) δ-toxin fibrils (pH 10). Please note that the scale bar changes.

## Data Availability

The data presented in this study are openly available in FigShare. https://doi.org/10.6084/m9.figshare.13553543.v1

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
