# Peer review of "Modulating Kinetics of the Amyloid-Like Aggregation of S. aureus Phenol-Soluble Modulins by Changes in pH"

_microorganisms, 2021, doi:10.3390/microorganisms9010117_

Round 1
Reviewer 1 Report
I think it is overall an interesting paper providing an insight into the aggregation properties of S. aureus PSM peptides in vitro. Data are interesting and worth publishing. However, a few issues need to addressed before the paper could be accepted.
Major issue.
- Authors term PSM aggregates amyloids, however the evidence for that is not obvious. True, ThT binding is a characteristic of amyloids (but is not specific only to them). CD and FTIR spectra only show conformational differences, however they per se can’t prove amyloid structure; microscopy identifies fiblils, but not all fibrils are amyloids. Are aggregates resistant to detergents? Did authors try binding to Congo Red with birefringence in polarized light? These features would make identification of fibrils as amyloids stronger.
Other issues.
- There are many examples of amyloids regulated by pH, for example Pmel17, insulin, some derivatives of synuclein, some artificial peptides, More detailed discussion of literature data on pH dependence of amyloid formation would be appropriate.
- Lines 36-38: S. cerevisiae Mot3 can not be an example of “functional bacterial amyloid”, as ueast is a eukaryotic microorganism (fungus). Note also that Mot3 does not form biofilms by itself, it is an intracellular protein regulating expression of the genes coding for proteins involved in cell wall and filamentous growth; its inactivation via amyloid formation modulates biofilms indirectly. In the context of the paper, it could be more relevant to mention fungal hydrophobins.
Author Response
Response to Reviewer #1s’ comments
We highly appreciate the positive comments provided by the Reviewers that greatly contributed to the manuscript and clarified the results. Below, please find a point-by point response to the Reviewers’ comments. Main changes made in the manuscript are highlighted in the revised version.
Reviewer #1
Comments and Suggestions for Authors
I think it is overall an interesting paper providing an insight into the aggregation properties of S. aureus PSM peptides in vitro. Data are interesting and worth publishing. However, a few issues need to addressed before the paper could be accepted.
Major issue.
- Authors term PSM aggregates amyloids, however the evidence for that is not obvious. True, ThT binding is a characteristic of amyloids (but is not specific only to them). CD and FTIR spectra only show conformational differences, however they per se can’t prove amyloid structure; microscopy identifies fiblils, but not all fibrils are amyloids. Are aggregates resistant to detergents? Did authors try binding to Congo Red with birefringence in polarized light? These features would make identification of fibrils as amyloids stronger.
Response: We agree with the reviewer that ThT binding is a characteristic of amyloids but not specific only with them as some literatures also report that they bind to amorphous aggregates too. In our study to characterize the aggregates, we executed the secondary structural characterization by CD and FTIR. It is reported in literature that double minima at 208 and 222 nm are indicative of α-helical structure in PSMs (Da et al., Front Cell Infect Microbiol. 2017; 7: 206.). All PSMs in their native state i.e., prior to aggregation show double minima at 208 and 222 nm indicative of α-helical structure (Zaman M et al., eLife Dec.2020. https://elifesciences.org/articles/59776). However, when incubated with different pH for various time intervals, the observed spectra exhibited characteristic minimum shifting at ~218 nm for PSMα1, PSMα4 and δ-toxin and ~220 nm for PSMβ1 and PSMβ2, which clearly indicates strong β-sheet signatures as reported in many previous literatures (Dueholm et al, Biochemistry. 2011 Oct 4; 50(39): 8281–8290, Marnelle et al., 2016, Scientific report, 6, 34552). Further, the observed peak for PSMα3 at 208 nm and 228 nm indicates that it forms α-helical like amyloidal structure and are in good agreement with recent finding (Tayeb-Fligelman et al., 2017, Science, 355, 831-833). Consistent with SRCD findings, FTIR also showed very similar spectra for the fibrils with a well-defined peak at around 1625 cm−1 for all PSMs except PSMα3, indicative of β-sheet in an amyloid-like conformation maintained by very strong hydrogen bonds (Dueholm et al, Biochemistry. 2011 Oct 4; 50(39): 8281–8290,). However, we observe the intensity of peaks for different PSMs at various pH, which are indicative of aggregation propensity for PSMs.
Further, we agree with the reviewer that our microscopic analysis shows that not all fibrils are amyloid. We reported that we observe some fibrils and some amorphous aggregates with solution conditions. We have reported that PSMα4 and PSMβ1 show heterogeneity in the fibrillar structure as it forms fibrils and amorphous aggregates at acidic conditions. Still many works are going on to characterize the individual fibrils of PSMs at their atomic level. In contrast to β-sheet like structure, PSMα3 forms cross α-like amyloidal structure as reported by Fligelman et al 2017, Science, 355, 831-833. In this study, we do not do experiments for the fibrils stability with chemical denaturants. However, we recently showed that all fibrils except PSMα3 are thermally stable and resistant to higher concentration of chemical denaturants (Zaman M et al., eLife Dec.2020 https://elifesciences.org/articles/59776). Further, we have not performed congo red binding with birefringence in polarized light. We have reported the comparative study of PSMs at different solution conditions and unfortunately, CR assay could not be performed below pH 5.0 because CR changes color under such conditions (Khan et al Soft Matter, 2014 Apr 21; 10(15):2591-9).
Other issues.
- There are many examples of amyloids regulated by pH, for example Pmel17, insulin, some derivatives of synuclein, some artificial peptides, More detailed discussion of literature data on pH dependence of amyloid formation would be appropriate.
Response: As per you valuable suggestions, we have included all said works regulated by pH in our revised manuscript to make our literature more appropriate.
- Lines 36-38: S. cerevisiae Mot3 cannot be an example of “functional bacterial amyloid”, as yeast is a eukaryotic microorganism (fungus). Note also that Mot3 does not form biofilms by itself, it is an intracellular protein regulating expression of the genes coding for proteins involved in cell wall and filamentous growth; its inactivation via amyloid formation modulates biofilms indirectly. In the context of the paper, it could be more relevant to mention fungal hydrophobins.
Response: We apologize for the mistake and corrected in our revised manuscript. On reviewer’s suggestion, we have modified the context of paper and incorporated the suggestions.
Reviewer 2 Report
Title: Modulating kinetics of amyloid formation of S. aureus phenol soluble modulins by changes in pH.
This manuscript describes aggregation behavior of a series of peptides in S. aureus called phenol soluble modulins (PSMs) in response to changes in pH. These peptides are implicated in forming and stabilizing biofilms in S. aureus as well as other pathogens with relevance toward persistence and virulence. The authors use fluorescence, CD, FTIR and TEM to characterize fibril formation and structure across all 7 PSMs encoded in S. aureus. The results indicate some pH dependence of aggregation kinetics and some differences in amyloid structures of the different PSMs. The study is fairly straightforward, and the experiments well executed for the most part. However, there are some areas that require clarification and a number of minor errors that must be corrected as indicated below:
Major Revisions:
- There appear to be some significant differences between the aggregation kinetics observed in this study and those on the alpha PSMs in reference 11 using the same fluorescence assay. Namely, in the previous work, alpha-4 was observed to exhibit the most rapid aggregation kinetics followed by alpha-1 while the other alpha PSMs or delta toxin did not appear to form fibrils. This is very different from what is observed in the current work where alpha-4 does not appear to aggregate above pH 3. The authors should offer some explanation of these differences.
- There are rather dramatic differences in the fluorescence intensities in Figures 1 and 2. Is this indicative of the degree of amyloid formation? If so, it should be described. Some samples are only recorded for a fraction of the time (e.g. alpha-3 at pH 7). Why?
- The identities of various buffers are given in the materials and methods, but their concentrations are not. Was any effort made to normalize the ionic strength at each pH? This is a confounding variable if not, as ionic strength has been observed to significantly impact the process (reference 22).
- It would be helpful to include a supplementary figure with the sequences of the PSMs.
- The rationale of the work is not clearly articulated. Are there a large range of pH values for environments in which S. aureus forms biofilms? A description of this in the introduction and/or discussion would be helpful.
Minor Revisions: I have pointed out some specific corrections here, but the manuscript needs to be proofed for correct English grammar.
- Line 41, “structure” should be “structures”
- Line 46, delete “of”
- Line 50, “does not” should be “do not”. This statement also indicates that alpha 4 does not aggregate, which is in contrast to what is observed in reference 11 (see point 1 above). Finally, the word “quiescent” is used and here and throughout the paper. What are “quiescent conditions” as the term means dormant or inactive.
- Line 41, “kinetic” should be “kinetics”
- Line 173, should be “at increased pH values”
- Line 203, “curved” should be “curves”
- Line 217, delete “is”
- Line 228, “charged” should be “charge”
- Line 239, “form” should be “from
- Lines 288-289, the statement ”the reported cross-alpha-helical structure...” requires a reference.
- Line 295, “acidic” should be “basic”
- Line 365, “Transmission” does not need to be capitalized.
- Line 369, should be “a heterogenous species appears”
- Line 386, “aggregated” should be “aggregates”
- Line 413, it is unclear what the authors are trying to say. Should this read “that different PSM peptides...”?
- Line 415, “exhibits” should be “exhibit”
- Line 423, “Despite of” should be “despite the”
- Lines 430-431: Again this is unclear. Consider “indicates that despite the observation that higher pH inhibits biofilm maturation, PSM peptides formed functional amyloids at higher pH.”
- Lines 432-433: The data in this manuscript do not support this statement as the contribution of each peptide to biofilm formation was not evaluated. It this is coming from previous work, it should be referenced.
Author Response
Response to Reviewer #2s’ comments
We highly appreciate the positive comments provided by the Reviewers that greatly contributed to the manuscript and clarified the results. Below, please find a point-by point response to the Reviewers’ comments. Main changes made in the manuscript are highlighted in the revised version.
Reviewer #2
This manuscript describes aggregation behavior of a series of peptides in S. aureus called phenol soluble modulins (PSMs) in response to changes in pH. These peptides are implicated in forming and stabilizing biofilms in S. aureus as well as other pathogens with relevance toward persistence and virulence. The authors use fluorescence, CD, FTIR and TEM to characterize fibril formation and structure across all 7 PSMs encoded in S. aureus. The results indicate some pH dependence of aggregation kinetics and some differences in amyloid structures of the different PSMs. The study is fairly straight forward, and the experiments well executed for the most part. However, there are some areas that require clarification and a number of minor errors that must be corrected as indicated below:
Major Revisions:
- There appear to be some significant differences between the aggregation kinetics observed in this study and those on the alpha PSMs in reference 11 using the same fluorescence assay. Namely, in the previous work, alpha-4 was observed to exhibit the most rapid aggregation kinetics followed by alpha-1 while the other alpha PSMs or delta toxin did not appear to form fibrils. This is very different from what is observed in the current work where alpha-4 does not appear to aggregate above pH 3. The authors should offer some explanation of these differences.
Response: We agree with the reviewer that significant differences between aggregation kinetics were observed for different PSMs. With reference to the findings presented in reference 11, it is reported that only PSMα1 and PSMα4 show aggregation behavior while the rest of the peptides do not aggregate. However, we have recently reported the aggregation behavior and molecular mechanism of aggregation for the seven peptides individually (Zaman M et al., eLife Dec.2020 https://elifesciences.org/articles/59776). In that report we found that except for PSMα2 and δ-toxin all peptides aggregates under quiescent conditions. We also observed the fast aggregation and elongation rate of PSMα3 in comparison to other peptides. However, the work presented here is very different from the previous one. In this, we observe how aggregation kinetics changes with change in solution conditions. δ-toxin, which was not showing aggregation behavior at high monomeric concentration, produces fibrillary structure at acidic and basic pH conditions. We have also shown that PSMα4, which was not showing sigmoidal behavior of kinetics, produces sigmoidal kinetics at acidic conditions. Although, we do not observe sigmoidal kinetics for PSMα4 at basic conditions, but they tends to aggregate and produces nice fibrillary structure as confirmed by SRCD, FTIR and microscopic analysis. Further, we have also reported that how solution conditions changes the lag times of PSMs, which provide structural integrity to biofilms.
- There are rather dramatic differences in the fluorescence intensities in Figures 1 and 2. Is this indicative of the degree of amyloid formation? If so, it should be described. Some samples are only recorded for a fraction of the time (e.g. alpha-3 at pH 7). Why?
Response: We observed dramatic differences in the fluorescence intensities in figure 1 and 2 for different PSM peptides. Yes, this may be due to degree of amyloid formation as ThT binding is more prominent. We have included this in our revised manuscript. Out of the seven peptides, PSMα3 aggregates very fast and attain a stationary phase within one hour hence completing the aggregation reaction. For this reason we only measure the kinetics for PSMa3 for a fraction of the time as compared to the other PSM peptides that require longer incubation time to complete the aggregation reaction.
- The identities of various buffers are given in the materials and methods, but their concentrations are not. Was any effort made to normalize the ionic strength at each pH? This is a confounding variable if not, as ionic strength has been observed to significantly impact the process (reference 22).
Response: We agree with the reviewer that the ionic strength of the solution can impact the aggregation reaction. We have added the concentration of the buffers (20 mM) in the materials and methods section. Since the concentration of the buffers was kept the same throughout the experiments the ionic strength doesn’t vary.
- It would be helpful to include a supplementary figure with the sequences of the PSMs.
Response: As per reviewer’s suggestion, we had incorporated a table showing the peptide sequence of individual PSMs in our revised manuscript (Sup.TableS4).
- The rationale of the work is not clearly articulated. Are there a large range of pH values for environments in which S. aureus forms biofilms? A description of this in the introduction and/or discussion would be helpful.
Response: Biofilms are known to affect wound healing in chronic wounds and their ability to eradication is significantly affected by changes in pH. That is why we have seen the effect of pH on the formation of functional amyloids by PSMs, which are considered to be associated with the stabilization of biofilm formation. We have included these points in our introduction section. Further, previous literatures clearly states that biofilm formation is largely affected by the pH levels (Zmantar T, et al New Microbiol 2010; 33:137–45 Chaieb K et al. Annals of Microbiology. 2007,7, 31-437). As per your suggestion, we have included these findings in the introduction section in our revised manuscript to make this point more clear for readers.
- Minor Revisions: I have pointed out some specific corrections here, but the manuscript needs to be proofed for correct English grammar.
Response: We have sufficiently improve the grammar and English to enhance overall readability of the manuscript as per reviewer’s suggestion.
- Line 41, “structure” should be “structures”
Response: We have corrected it in our revised manuscript.
- Line 46, delete “of”
Response: We have corrected it in our revised manuscript.
- Line 50, “does not” should be “do not”. This statement also indicates that alpha 4 does not aggregate, which is in contrast to what is observed in reference 11 (see point 1 above). Finally, the word “quiescent” is used and here and throughout the paper. What are “quiescent conditions” as the term means dormant or inactive.
Response: The statement that PSMα4 do not show aggregation means it is not showing sigmoidal aggregation kinetics under the conditions that we have used. In our recent report, an increase in ThT fluorescence was observed for PSMα4 along with β-sheet like structures and fibrillar morphology (Zaman M et al., eLife Dec.2020 https://elifesciences.org/articles/59776). That supports the previous finding which we have been mentioned in our reference. However, in this report we observe sigmoidal curve kinetics for PSMα4 under acidic and basic conditions. Further, Salinas et al. (Nature Communications 9, 3512 (2018)) also observed kinetics of PSMα4 under shaking conditions and a lag time of 20 h was observed which is different to that of Marnelle et al work (reference 11), who observed fibrillar structure after six days. These findings clearly supports that PSMα4 show different aggregation behavior at various conditions. This might be due their hydrophobic nature, which greatly influence the amyloid formation of peptides. We have done our experiments under “quiescent conditions” which means we do not shake or put any facilitating agent in the solutions. We have added this information in the materials and methods section to make it clear to the reader.
- Line 41, “kinetic” should be “kinetics”
Response: We have corrected it in our revised manuscript.
- Line 173, should be “at increased pH values”
Response: We have corrected it in our revised manuscript.
- Line 203, “curved” should be “curves”
Response: We have corrected it in our revised manuscript.
- Line 217, delete “is”
Response: We have corrected it in our revised manuscript.
- Line 228, “charged” should be “charge”
Response: We have corrected it in our revised manuscript.
- Line 239, “form” should be “from
Response: We have corrected it in our revised manuscript.
- Lines 288-289, the statement ”the reported cross-alpha-helical structure...” requires a reference.
Response: As per reviewer’s suggestion, we have incorporated a reference to support our statement.
- Line 295, “acidic” should be “basic”
Response: We have corrected it in our revised manuscript.
- Line 365, “Transmission” does not need to be capitalized.
Response: We have corrected it in our revised manuscript.
- Line 369, should be “a heterogenous species appears”
Response: We have corrected it in our revised manuscript.
- Line 386, “aggregated” should be “aggregates”
Response: We have corrected it in our revised manuscript.
- Line 413, it is unclear what the authors are trying to say. Should this read “that different PSM peptides...”?
Response: We apologize for the mistake and corrected in our revised manuscript. Yes, we are trying to say that different PSM peptides ensure formation of functional amyloids at a large range of pH values.
- Line 415, “exhibits” should be “exhibit”
Response: We have corrected it in our revised manuscript.
- Line 423, “Despite of” should be “despite the”
Response: We have corrected it in our revised manuscript.
- Lines 430-431: Again, this is unclear. Consider “indicates that despite the observation that higher pH inhibits biofilm maturation, PSM peptides formed functional amyloids at higher pH.”
Response: We apologizes for the sentence, which is unclear for the readers. We have corrected it for better understanding in our revised manuscript.
- Lines 432-433: The data in this manuscript do not support this statement, as the contribution of each peptide to biofilm formation was not evaluated. It this is coming from previous work, it should be referenced.
Response: In this statement, we meant to say that nearly physiological pH, its β-group of peptides i.e. PSMβ1 and PSMβ2, which forms functional amyloids along with PSMα3. It is well known in previous literature (Schwartz, et al 2012 Plos pathogen) that PSMs functional amyloids play significant role in sustaining the integrity of biofilms in Staphylococcus aureus. We have incorporated the said reference in our revised manuscript to improve clarity.
Round 2
Reviewer 1 Report
While authors addressed some of comments, I still have doubts regarding amyloid nature of the aggregates described in this paper. As per authors response, the most stringent assay (CR binding with birefringence) is not working at low pH. However, all other assays do not provide a direct evidence; while data are consistent with the amyloid nature of fibrils, they don't prove amyloid nature. References to previous work by authors don't provide sufficient direct evidence either. Resistance to denaturing agents was observed not for all fibrils (and on the other hand, strong denaturing agents may solubilize some "classic" amyloids as well, while some other types of aggregates could stay resistant). At the very least, rechecking resistance to mild detergents such as SDS or sarxosyl, possibly with subsequent characterization of polymers by using agarose (SDD-AGE, see Kushnirov et al. 2003 J Biol Chem for example) and/or native acrylamide gels would be highly desirable. Alternatively, authors may change wording throughot the paper and talk about aggregates/fibrils with some characteristics of amyloids, not insisting that they are proven amyloids; this would apply to both title and abstract as well.
Another issue that still needs to be addressed is the statement of pH dependence of amyloid formation being "highly unusual" (starting from Abstract), that ignores previous work with some eukaryotic amyloidogenic proteins such as Pmel17, insulin and even some synuclein-derived constructs, where a pattern of pH dependence is clearly established and well studied.
Author Response
In accordance with the comments of the reviewer we have changed the wording throughout the manuscript. The aggregates are no longer called amyloids in the manuscript but rather aggregates and fibrils with amyloid-like characteristics. This has also been changed in the title and the abstract.
We have also removed the claim that the pH dependence of amyloid formation as being “highly unusual” from the manuscript. We apologize for not correcting this in the previous round of revisions as it should have been removed after we incorporated more references as suggested by the reviewer.
Reviewer 2 Report
The revised manuscript is significantly improved in terms of experimental descriptions and readability. The authors have responded to each of my concerns adequately save one. It is still not clear how constant ionic strength was maintained in all pH conditions. The authors state that all buffers were used at 20 mM. However, that does not indicate that the total ion concentrations are all the same. In fact, it seems impossible that a HCl solution at pH = 1.0 can be 20 mM. The protonation states of all species at a given pH, their concentrations and the concentrations of counterions must all be taken into account. If this calculation has been performed and ionic strength in each buffer maintained, this should be clearly stated. If not, it is not necessarily a critical flaw. However, the ionic strength of each buffer solution should be calculated and reported, and the claim that ionic strength was equivalent across all conditions must be removed.
Author Response
We appreciate the reviewer’s comment and suggestion. Additionally, we apologizes for the unclear statement in the materials and method section, especially for the ionic concentration of different buffers. Earlier we have only reported the molarity of all the buffers. We have now added the actual calculated ionic strength of each individual buffer in our revised manuscript that have been used in our study for the aggregation of PSMs peptides. Further, we have also removed the claim that ionic strength was equivalent across all conditions.
Round 3
Reviewer 2 Report
I appreciate that the indication that ionic strength was normalized across all conditions has been removed. However, I still do not understand how the authors can arrive at an ionic strength for HCl at pH=1 of ~0.02. By definition, the proton concentration of such a solution is 100 mM, so the chloride concentration must be at least that high. It seems to me that the ionic strength must be at least 0.1. Can the authors indicate how ionic strength was calculated? Although none of this appears to affect the conclusions of the work, it still needs to be correct.
Author Response
There was a typo in the ionic strength for the pH1 solution it was 0.109 and not 0.019. This has been corrected and the other ionic strengths has been double checked.